# Correlation between Heart fatty acid binding protein and severe COVID-19: A case-control study

**Li Yin[1], Huaming Mou[1], Jiang Shao[1], Ye Zhu[2], Xiaohua Pang[1], Jianjun Yang[1], Jianming Zhang[1], Wei Shi[1], Shimei Yu[3], Hailong Wang[1] ***

**1** Department of Cardiology, Heart Center, Chongqing Three Gorges Central Hospital, Affiliated Three Gorges Hospital of Chongqing University, Chongqing, China, **2** Department of Cardiology, West China Hospital of Sichuan University, Chengdu, China, **3** Department of Otorhinolaryngology, Wanzhou First People Hospital, Chongqing, China

๑ These authors contributed equally to this work.

* wangjinlong2007666@163.com

**Data Availability Statement:** All relevant data are within the manuscript and its Supporting Information files.

**Funding:** Unfunded studies.

## Abstract

### Background

Heart-fatty acid binding protein (HFABP) has been recognized as a highly heart-specific marker. However, it is currently unknown that its HFABP is also closely related to the severity of COVID-19.

### Methods

We retrospectively screened 46 patients who met our inclusion criteria within 4 weeks. They were tested for HFABP after the diagnosis of COVID-19, and monitored for HFABP during their hospital stay. We tracked the patients during their hospital stay to determine if they had severe COVID-19 or mild-to-severe transition features. We calculated the chi-square test values found for HFABP to predict the correlation between HFABP levels and the severity of the COVID-19.

### Results

Of these 46 cases, 16 cases with confirmed COVID-19 were tested for HFABP> 7 ng / mL upon admission; among them, 14 cases were diagnosed with severe COVID-19 within the hospitalization. The Odds ratio of the measured HFABP elevation was 6.81(95% confidence interval [CI] 5.23–8.40), and 3 patients with severe COVID-19 progressed in 5 patients with mild HFABP> 7 ng/mL.

### Conclusion

These data indicate that the elevation of HFABP is closely related to the severity of COVID-19 in the patients, and the elevated HFABP may cause rapid development of patients with mild COVID-19 into severe COVID-19. But serum HFABP negative maybe make patients with mild COVID-19 safer, the current data show no effect on the all-cause mortality.

**Competing interests:** No authors have competing interests.

## Trial registration

Our study has been registered with the Chinese Clinical Trial Registry, the registration number: ChiCTR2000029829.

## Introduction

Coronaviruses belong to the Coronaviridae family of non-segmented positive-sense RNA viruses and are widely parasitic in humans and other mammals [1]. Although most infections with human coronavirus are mild, two coronaviruses, including severe acute respiratory syndrome coronavirus (SARS-CoV) [2–4] and Middle East respiratory syndrome coronavirus (MERS-CoV) [5,6], cause severe infection. They can cause fulminant disease and severe illness. In December 2019, a new coronavirus named 2019 New Coronavirus (2019-nCoV) was found in Wuhan, Hubei, China. The disease caused by this coronavirus is COVID-19 [7–11]. At present, many cases have been confirmed in all provinces of China and in other countries. For making the diagnosis of patients with COVID-19, it is a great challenge for doctors to determine the condition of patients with severe illness as early as possible.

Heart fatty acid-binding protein, a serum biomarker for myocardial injury, is highly cardiac specific [12–14]. Recently, we have found that elevated HFABP levels are associated with severe COVID-19 or mild-to-severe transition features. Our study sought to determine whether the measurement of the HFABP can predict short-term turnover and prognosis in patients with COVID-19.

Our study retrospectively analyzed the epidemiological, clinical, and laboratory characteristics of patients with COVID-19 and compared HFABP levels with severe COVID-19 and mild-to-severe transition features. We hope that our findings will provide information to the global community about predicting the condition and outcomes of patients with COVID-19.

## Materials and methods

### Patients and trial designs

During January 2020, we retrospectively screened all patients with COVID-19 admitted to Chongqing Three Gorges Central Hospital, and we obtained approval from the Ethics Committee of the Chongqing Three Gorges Central Hospital. Eligible patients included patients older than 14 years of age, patients who were diagnosed with COVID-19, and patients who had been assessed for HFABP serum concentrations at any time during the hospital stay. We excluded patients who were younger than 14 years of age, patients who had not received the measurement of HFABP serum concentrations at any time during the hospital stay. We performed follow-up (during the hospital stay, 9–21 days) of the patients in the study and recorded their status (mild, severe, or death) and whether they changed from mild to severe, defined as any of the following conditions:

Mild COVID-19: The patient presents with only fever, respiratory tract infection, and other symptoms, and imaging shows pneumonia. Those who have one of the following pathogenic evidences: 1. Real-time fluorescent RT-PCR of respiratory specimens or blood specimens for detection of novel coronavirus nucleic acid; 2. Sequencing of viral genes of respiratory specimens or blood specimens, highly homologous to known novel coronavirus.

Severe COVID-19: confirmed as COVID-19 and meets any of the following criteria: 1. Respiratory distress, RR $\geq$ 30 times / min; 2. Means oxygen saturation $\leq$ 93%; Arterial blood oxygen partial pressure (PaO2) / oxygen concentration (FiO2) $\leq$ 300 mmHg (1 mmHg = 0.133 kPa).

Death: Total deaths from all causes during the hospital stay.

## Measurement

All blood samples were collected and sent to the laboratory immediately. All selected patients were measured for HFABP. The results were provided to the clinical medical staff. HFABP levels were measured by the Roche Modular Analyzer (Roche Diagnostics, Laval, Quebec). These tests were performed with reagents provided by the manufacturer and in strict accordance with the procedures. The cut-off value is defined as the internationally agreed value of 7 ng / mL.

## Statistical analyses

Categorical data were expressed as counts and percentages. Continuous data for normal and skew distribution are expressed as mean standard deviation and median, respectively. The Kolmogorov Smirnov test was used to test the normality of the data distribution. Categorical variables are expressed as numbers (%) and compared between the HFABP-raised group and the normal group by the $Chi^2$ test or the Fisher's exact test, and classification clinical, characteristics, and outcome rates were tested using the $Chi^2$ test.

A $P$ value of less than 0.05 was considered statistically significant. Statistical analysis was performed using SPSS software(version 23, IBM Inc., Armonk, NY, USA).

## Result

### Patients

During the research period, 245 patients arrived at our hospital and were diagnosed with COVID-19. Of these patients, 199 patients were excluded from our research because they did not undergo testing for serum HFABP within a week of admission, and 46patients (25 patients with severe disease and 21 patients with mild disease) were finally included. In our study, 45 patients were tested for serum HFABP from the day of admission to the 5th day of admission, and one patient was tested for serum HFABP on the 6th day of admission. In addition, 2 critically severe COVID-19 patients died during hospitalization. Three COVID-19 patients with serum HFABP-positive result changed from mild to severe state during hospitalization. Demographic and disease characteristics of 15 patients with serum HFABP-positive COVID-19 and 30 patients with serum HFABP- negative COVID-19 are presented in Table 1. The results of HFABP analysis showed a normal distribution, with values decreasing from 1.76 ng/mL to 24.68 ng/mL (mean 6.81, 95% CI 5.23–8.40, SD 5.33).

### Outcomes

Table 2 summarizes the relationship between positive HFABP and severe COVID-19. In the HFABP positive group, a significant increase in the prevalence of severe illness was observed during hospitalization of patients with COVID-19 (87.5% vs 40%, P = 0.002). Among them, in the HFABP positive group, 3 of 5 patients with mild COVID-19 worsened to severe COVID-19 during hospitalization, and the incidence was 60%. One patient died in each of the two groups (P > 0.05), which was not statistically significant (Table 2).

**Table 1. Demographic and disease characteristics of enrolled patients.**

| Characteristic | Group; no. (%) of patients* | | |
|---|---|---|---|
| | All n = 45 | HFABP Positive n = 15 | HFABP negative n = 30 |
| Male | 25 | 5 | 22 |
| Age, mean (SD), yr | 52.4 | 65.3(17.9) | 45.6(12.5) |
| Tobacco smoking | 3 | 0(0) | 3(9.6) |
| Car T | 39 | 3(7.6) | 2(5.1) |
| **Any comorbidity** | | | |
| Diabetes | 7 | 2(14.2) | 5(16.1) |
| Hypertension | 4 | 1(7.1) | 3(9.6) |
| Cardiovascular disease | 2 | 1(7.1) | 1(3.2) |
| Malignancy | 3 | 1(7.1) | 2(6.4) |
| COPD | 2 | 0(0) | 2(6.4) |
| CLD | 0 | 0(0) | 0(0) |
| CKD | 0 | 0(0) | 0(0) |

Chronic obstructive pulmonary disease = COPD, Chronic liver disease = CLD, Chronic kidney disease = CKD, Cardiac troponin T = Ca T. Note: All differences were statistically nonsignificant. SD = standard deviation.
*Except as indicated for Age.

## Discussion

Both SARS-CoV and MERS-CoV are thought to originate in bats, and many studies have found coronaviruses with many other genomic sequences in bats. In 2013, Ge and colleagues reported the genome-wide sequence of a new coronavirus similar to SARS in bats. This virus can utilize human receptors and has the potential to replicate in human cells. 2019-nCoV has the potential to cause a pandemic, and it is still being studied in depth to prevent it from becoming a global health threat. Reliable and rapid differential diagnosis and reasonable treatment of diagnosed patients with COVID-19 are still essential to control the epidemic [15,16]. Our study screened 46 laboratory-confirmed patients with COVID-19. The patient was severe viral pneumonia and was fatal. All patients were sent to Chongqing Three Gorges Central Hospital before February 22, 2020, and their clinical symptoms were very similar to SARS. Acute respiratory distress syndrome (ARDS) can occur in severe patients, and they will require admission to the Intensive Care Unit and assistant treatment with mechanical ventilation. During the retrospective study, 2 of 46 patients included in this research died (4.3%); thus, the mortality of COVID-19 was very high.

At present, the determination of severe COVID-19 is mainly based on the comprehensive analysis of clinical symptoms, signs, and blood gas analysis results. The global judgment of the severity of COVID-19 is based on the Chinese guidelines, and there is no clinical serum marker for comprehensive judgment. This study shows that in patients with COVID-19,

**Table 2. Clinical outcomes of the matched study population of COVID-19 with HFABP levels.**

| Clinical outcomes | HFABP Positive group N = 15 | HFABP Negative group N = 30 | P value |
|---|---|---|---|
| **Severe COVID-19** | 13 | 12 | 0.002 |
| **Mild-to-severe COVID-19** | 0 | 3 | / |
| **Death** | 1 | 1 | >0.05 |

Heart fatty acid-binding protein = HFABP.

elevated serum HFABP is closely related to the severity of disease in the patients, and there is a significant statistical difference from patients with normal serum HFABP. Therefore, elevated serum HFABP can be used as an indicator of severe COVID-19 and an independent risk factor for patient prognosis. Among the patients in this study, 40 patients were monitored for troponin T, and only 6 patients were positive for troponin T; however, there was no statistical difference between the serum HFABP-negative and HFABP-positive groups. The hypothesis that HFABP is affected by troponin T does not hold, and the serum HFABP levels show an independent stable performance. This can happen because, like SARS-CoV and MERS-CoV, novel coronavirus infections also induce the secretion of a large number of cytokines [17–21], leading to inflammatory lung injury, which reduces blood oxygen concentration and puts myocardial cells in a hypoxic state, thereby increasing the release of HFABP into the blood. We noted that 86.7% of patients with elevated serum HFABP are patients with severe COVID-19; therefore, it is of great significance to judge and predict the outcome of patients with severe COVID-19. At present, there is no objective laboratory index for assessing the outcome of patients with COVID-19. Serum HFABP can be used as an effective index; thus, it can guide the clinicians to judge patients with severe COVID-19 in the short term, and elevated HFABP can also severely affect the outcome of patients with COVID-19.

We also observed a phenomenon in which five patients with mild COVID-19 were positive for serum HFABP upon admission. Three patients deteriorated to severe neo-coronavirus pneumonia very quickly after admission, and the incidence of this event was 60%. However, 30 patients with mild COVID-19 were negative for serum HFABP, and none of these patients developed severe disease. Therefore, we speculate that if patients with mild COVID-19 are positive for serum HFABP, they can easily deteriorate to severe COVID-19. Based on the small sample size of our screened cases, we did not perform a statistically significant difference analysis.

In addition, two deaths were reported in our study, and they were severe COVID-19 patients. There was one case in the HFABP positive group, and the other case was in the HFABP negative group. We performed a statistical analysis for these groups, and there was no statistical difference in mortality between the two groups. However, we avoided sensitivity and specificity analysis because the sample size was small and we could not draw convincing conclusions. A larger sample size is needed for further confirmation.

## Conclusion

Our conclusion is that the elevation of HFABP is closely related to the severity of COVID-19 in the patients, and the elevated HFABP may cause rapid development of patients with mild COVID-19 into severe COVID-19. But serum HFABP negative maybe make patients with mild COVID-19 safer, the current data show no effect on the all-cause mortality. New coronaviruses have acquired effective human transmission capabilities [17,22]. We are deeply aware of the challenges and concerns that our global medical care encounters with COVID-19. Every effort should be made to study and control this disease. Therefore, we recommend worldwide promotion of HFABP application, which will help to judge and predict severe COVID-19.

## Supporting information

**S1 Data.**
(XLSX)

**S2 Data.**
(XLSX)

## Acknowledgments

We thank LetPub (www.letpub.com) for its linguistic assistance during the preparation of this manuscript.

## Author Contributions

**Conceptualization:** Li Yin, Jiang Shao, Ye Zhu, Hailong Wang.

**Data curation:** Li Yin, Jiang Shao, Hailong Wang.

**Formal analysis:** Li Yin, Huaming Mou, Ye Zhu, Jianming Zhang, Wei Shi, Shimei Yu, Hailong Wang.

**Funding acquisition:** Li Yin, Hailong Wang.

**Investigation:** Xiaohua Pang, Jianjun Yang, Jianming Zhang.

**Methodology:** Li Yin, Huaming Mou, Jiang Shao, Ye Zhu, Hailong Wang.

**Project administration:** Li Yin, Hailong Wang.

**Resources:** Huaming Mou.

**Software:** Jianming Zhang.

**Supervision:** Jiang Shao.

**Validation:** Huaming Mou.

**Visualization:** Jianming Zhang, Wei Shi, Shimei Yu.

**Writing – original draft:** Li Yin, Huaming Mou, Jiang Shao, Hailong Wang.

**Writing – review & editing:** Li Yin, Huaming Mou, Jiang Shao, Ye Zhu, Xiaohua Pang, Jianjun Yang, Jianming Zhang, Wei Shi, Shimei Yu, Hailong Wang.

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
