## [Decision Letter · Decision Letter 0]

31 Mar 2020

Correlation between H eart fatty acid binding protein and severe COVID-19 :A case-control study

PONE-D-20-05567

Dear Dr. Wang,

We are pleased to inform you that your manuscript has been judged scientifically suitable for publication and will be formally accepted for publication once it complies with all outstanding technical requirements.

With kind regards,

Xia Jin, MD, PhD

Academic Editor

PLOS ONE

Additional Editor Comments (optional):

Please add the references suggested by the reviewer to your revised ms.

Journal Requirements:

2. In ethics statement in the manuscript and in the online submission form, please provide additional information about the patient records/samples used in your retrospective study. Specifically, please ensure that you have discussed whether all data/samples were fully anonymized before you accessed them and/or whether the IRB or ethics committee waived the requirement for informed consent. If patients provided informed written consent to have data/samples from their medical records used in research, please include this information.

Reviewers' comments:

Reviewer's Responses to Questions

**Comments to the Author**

1. Is the manuscript technically sound, and do the data support the conclusions?

Reviewer #1: Yes

2. Has the statistical analysis been performed appropriately and rigorously? 

Reviewer #1: Yes

3. Have the authors made all data underlying the findings in their manuscript fully available?

Reviewer #1: Yes

4. Is the manuscript presented in an intelligible fashion and written in standard English?

Reviewer #1: Yes

5. Review Comments to the Author

Reviewer #1: This is a very timely, exciting and potentially transformative study of a potential blood biomarker for diagnosis of COVID-19 in symptomatic patients. The authors correctly state that currently severe COVID-19 is mainly diagnosed by clinical symptoms, and there is an urgent need for a serum marker that can be readily and rapidly performed. The study group patients were classified as mild or severe COVID according to the currently accepted pathological criteria. The focus on heart is justified because patients with heart disease are at higher risk for death and disability from the virus.

The novelty and high value of this study is that COVID patients were tested beginning with the day of admission for their serum level of heart fatty acid binding protein HFABP, an excellent choice for a candidate biomarker. H-FABP is an intracellular fatty acid binding protein found in high concentration in cardiac cells (myocytes). Its main function is to transport fatty acids inside the cell, and it does not normally leak though the cell membrane and enter the circulation. However, previous studies have extensively verified the discovery by the Glatz research group that after myocardial injury, damaged cells release HFABP. Is has been established as a marker for myocardial infarction, with higher sensitivity and specificity than the standard measure of troponin for an acute myocardial event and it is more rapidly released into the serum. The present study also measured troponin T in 40 patients, and only 6 showed elevated levels.

The current study is a milestone with its focus on the heart, and damage that can occur because of release of cytokines and by inflammatory responses. Moreover, previous studies have shown that the plasma concentration of HFABP has a prognostic value of death or MI after acute coronary syndrome one year. The current studies showed an association with higher levels of HFABP with severity of the COVID-19 in their patients. However, as a retrospective study, the elevated levels of HFABP cannot be a postulated to be a cause of the more rapid development.

Specific Comments:

Add refs below at top of p.3

Glatz JF, Kleine AH, van Nieuwenhoven FA, Hermens WT, van Dieijen-Visser MP, van der Vusse GJ (Feb 1994). "Fatty-acid-binding protein as a plasma marker for the estimation of myocardial infarct size in humans". British Heart Journal. 71 (2): 135–40. doi:10.1136/hrt.71.2.135. PMC 483632. PMID 8130020.

Kleine AH, Glatz JF, Van Nieuwenhoven FA, Van der Vusse GJ (Oct 1992). "Release of heart fatty acid-binding protein into plasma after acute myocardial infarction in man". Molecular and Cellular Biochemistry. 116 (1–2): 155–62. doi:10.1007/BF01270583. PMID 1480144.

Error in Table2: Table 2 has mislabeled the columns (HFABP negative should be positive)

6. PLOS authors have the option to publish the peer review history of their article (what does this mean?). If published, this will include your full peer review and any attached files.

Reviewer #1: Yes: James A. Hamilton

---

## [Editor Report · Acceptance letter]

21 Apr 2020

PONE-D-20-05567 

Correlation between H eart fatty acid binding protein and severe COVID-19 :A case-control study 

Dear Dr. Wang:

I am pleased to inform you that your manuscript has been deemed suitable for publication in PLOS ONE. Congratulations! Your manuscript is now with our production department. 

With kind regards,

on behalf of

Dr. Xia Jin 

Academic Editor

PLOS ONE